*forests*

*Article*

# Working Time, Productivity, and Cost of Felling in a Tropical Forest: A Case Study from Wijaya Sentosa's Forest Concession Area, West Papua, Indonesia

**Soenarno [1], Dulsalam [1], Yuniawati [1], Sona Suhartana [1], Seca Gandaseca [2,*], Yanto Rochmayanto [3], Achmad Supriadi [1] and Sarah Andini [1]**

1 Research Center for Biomass and Bioproducts, National Research and Innovation Agency-Republic of Indonesia, Jl. Raya Jakarta-Bogor KM. 46 Cibinong, Bogor 16911, Indonesia
2 Faculty of Agriculture, Universitas Nasional, Jalan Sawo Manila No. 61, Pejaten Barat, Pasar Minggu, South Jakarta 12520, Indonesia
3 Directorate of Environment, Maritime, Natural Resources, and Nuclear Policies, National Research and Innovation Agency-Republic of Indonesia, Gedung B.J. Habibie, Jalan Thamrin No. 8, Jakarta Pusat 10340, Indonesia
* Correspondence: secags66@gmail.com; Tel.: +62-812-8384-3672

**Abstract:** Felling of natural forest trees in West Papua Province is carried out mechanically using a chainsaw by applying a selective cutting silviculture system. This study aimed to determine the elements of working time, productivity, and cost of felling as well as factors influencing felling activities using the chainsaw in the tropical natural forests in West Papua, Indonesia. The felled trees are dominated by the meranti wood species group with a minimum diameter of 45 cm. The average volume of the felled tree is about 4.205 $m^3$/tree. The result showed that total felling time ranged from 15.3 to 18.3 min/tree with an average of 16.7 min/tree. The effective felling time was 11.8 min/tree (70.86%) and the delay time was 4.9 min/tree (29.14%). Felling time in making undercuts was on average 1.7 min/tree (9.98%), longer than back cuts time of 1.10 min/tree (6.39%). Total felling time was influenced by various variables: tree diameter, buttress height, and slope. Felling productivity was accounted for between 14.901 $m^3$/h and 17.067 $m^3$/h (15.778 $m^3$/h on average). Felling costs ranged from 3.366 USD/h to 3.473 USD/h with an average of 3.407 USD/h or equivalent to 0.209 USD/$m^3$ to 0.238 USD/$m^3$ with an average of 0.225 USD/$m^3$. To improve the effectiveness of felling time and productivity, this study suggests (1) upgrading the skills of the chainsaw operators through formal or in-house training in felling techniques, and (2) ensuring chainsaw operators bring a tree distribution map completed by field conditions information.

**Keywords:** tropical forest; West Papua; selective cutting; felling time; working time

check for updates

**Citation:** Soenarno; Dulsalam; Yuniawati; Suhartana, S.; Gandaseca, S.; Rochmayanto, Y.; Supriadi, A.; Andini, S. Working Time, Productivity, and Cost of Felling in a Tropical Forest: A Case Study from Wijaya Sentosa's Forest Concession Area, West Papua, Indonesia. *Forests* **2022**, *13*, 1789. https://doi.org/10.3390/f13111789

Academic Editor: Eduardo Tolosana

Received: 13 September 2022
Accepted: 27 October 2022
Published: 28 October 2022



## 1. Introduction

West Papua Province has a land area of 9.6 million hectares, around 87.3% (or an area of 8.39 million hectares) of which is a natural tropical forest. Timber harvesting activities are carried out using a selective felling system while the felling technique uses a reduced-impact logging approach to ensure forest sustainability. Felling consists of several elements, including walking among the trees, preparing the tree to be felled, determining the fall direction, making undercuts and back cuts, waiting for the tree to fall, stem measurement, and bucking. Felling is conducted by only cutting down commercial tree species that have been marked by an IDbarcode and registered on the tree distribution map.

In production forests, 40 cm is the minimum limit of the diameter (dbh) of trees that are allowed to be cut [1]. However, if the tree targeted to be cut has a major defect (for instance, hole diameter of ≥20 cm), the tree should not be cut, except for a special need, such as materials for constructing culverts or bridges. To maintain forest sustainability,

trees with a large diameter (>120 cm) should not be cut, because they are the source of seeds for generative reproduction [2].

Felling has the most important role affecting all subsequent harvesting stages [3]. The proper felling and bucking will affect wood quality, efficiency, and felling costs, thus affecting income from selling timber [4,5]. Therefore, felling activities require adequate skill to master felling techniques. Operator skills require at least three years after training to reach 100% of their potential productivity [6].

To date, chainsaws are the main tool for harvesting forests in the world due to their multifunctional uses and relatively low financial investment [7–9]. Chainsaws are widely used because of their ease of maintenance and operation. Another reason is that chainsaws provide high workability, to be able to increase efficiency and productivity, which leads to reduced felling costs. However, in practice, the use of chainsaws is limited to a few hours a day, it is in general used only for 1500–3000 h [10,11]. Felling trees using chainsaws consumes a lot of physical energy, causing chainsaw operators to get tired more quickly.

On the other hand, the process of tree felling is complex and involves many factors beyond human control [12]. Tropical natural forest usually has a heavy topography, slippery forest floor, dense tree stands, lianas entwined among the tree canopy, and rapidly changing weather. Bad weather can reduce efficiency and even stop felling operations completely [3]. Socio-economic conditions, forest conditions, slopes, work methods, and equipment used could also affect productivity and harvesting costs [13]. Such a condition is a heavy burden physically and psychologically, causing chainsaw operators to get tired quickly. Chainsaw operators must take frequent breaks to recover. The more frequent breaks, the greater the delay, and the working time will increase. As a result, it will reduce work productivity, and increase felling costs.

In addition, the logging concession company applies a full contractual wage system on felling activity based on the volume of log production. Unfortunately, a full contractual wage system encourages loggers (chainsaw operators) to work quickly to get the maximum volume of timber. Working in a hurry can result in improper felling operations. This leads to splitting the wood, damaging the tree, increasing delay time due to clamped saw, and even endangering the saw operator. Decreased productivity due to rushed work is a loss of wood volume of 11–18% [14].

It is necessary to conduct a work study on tree felling activity. A work study is one of the most common practices used worldwide to improve working productivity and performance [15,16]. According to Bjorheden [17], a work-study is applied as a means to investigate or improve production efficiency. Cutting trees using chainsaws is considered the most dangerous work in forest production activities and must be carried out by highly skilled and competent operators. In line with that, the study aims to analyze the effective working time, delay time, productivity, cost of felling using chainsaws, and factors affecting the process of felling to support harvesting effectivity in tropical natural forests. The results are expected to be useful for national and international implementation in the field and for making policy.

## 2. Materials and Methods

### *2.1. Study Site*

The research was carried out in 2018 in the forest concession area of Wijaya Sentosa's Concession located at Teluk Wondama Regency, West Papua Province, Indonesia. The condition of the research site was a logged-over area of dry land natural production forest. Timber species are dominated by group of meranti wood species such as meranti (*Shorea* spp.), resak (*Vatica papuana*), merbau (*Intsia* spp.), mersawa (*Anisoptera* sp.), merawan (*Hopea* sp.), nyatoh (*Palaquium* sp.), and matoa (*Pometia* spp.), and some are in the form of mixed types of forest wood species benuang (*Octomeles sumatrana*), bintangur (*Calophyllum* sp.), jambu-jambu (*Eugenia* spp.), mendarahan (*Myristica* spp.), kedondong hutan (*Spondias* spp.), and others. Felling is carried out with a selective cutting system with a minimum diameter limit of 40 cm. The topography of the research area includes sloping

to slightly steep with climatic conditions B according to Smith Ferguson. Soil types are alluvium formation, limestone formation, and clay rock formations.

The astronomical position of the research site was 134°16′–134°11′ east longitude and 3°35′–3°11′ south latitude as shown in Figure 1.

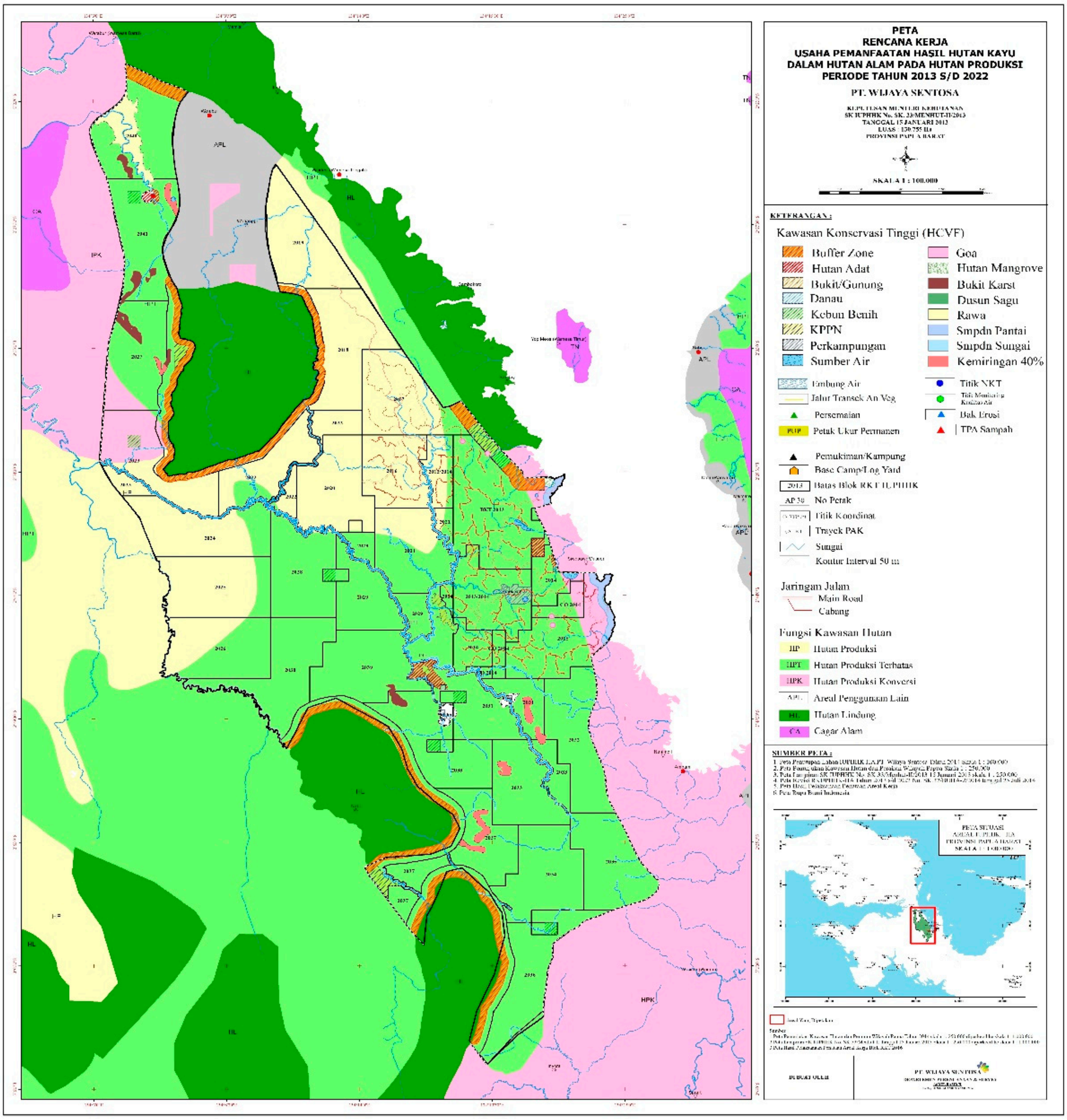


**Figure 1.** The position of the research site in the forest concession area of Wijaya Sentosa in West Papua, Indonesia.

## 2.2. Materials

The study used the following materials: a tally sheet (to record measurement results), a meter (to measure stump height, stem diameter, log length, and skidding length), a stopwatch (to measure the working time of felling activities), and a clinometer for measuring the slope of the forest land. The felling tool used was a chainsaw with a guide bar length of 70 cm and a power of 6.5 horsepower (HP). The chainsaw operator was about 35 years old and had never had any RIL training but he had more than 5 years of work experience. Work experience was gained by being self-taught as a chainsaw operator assistant.

### *2.3. Methods*

#### 2.3.1. Research Procedure

Field data collection was in the dry season but the forest soil conditions are humid. The research was conducted in the following stages:

1. Selected the distribution of sample plots representing the diversity of forest conditions. The study determined 3 sample plots on selected felling areas, each sample plot covering a 2 ha felling area (100 m × 200 m).
2. Sampling plots were distributed by a systematic approach with a random start, with a 50 m interval between the plots.
3. Conducted an inventory of targeted trees for felling (diameter of ≥40 cm) on each plot, measured the diameter at breast height (dbh), measured the branch-free stem height, and measure the buttressed height.
4. Tree felling and bucking.
5. Measured working time on each element of felling activities.

Tree preparation consisted of (1) determining the direction of falling trees, so the tree avoids falling into ditches, ravines/grooves, rocks, and other main trees; (2) cleaning the tree from the soil, sand, hard tree bark, etc. to prevent the chainsaw dulling quickly; (3) cleaning understorey and lianas propagating the felled trees; and (4) making an escape route for chainsaw operator. Tree preparation time depends on the tree diameter, number, shape, and height of the buttresses, the density of the understorey and liana around the tree, and the topography. Ideally, the escape route should be chosen at an angle of 45° behind the expected drop line and should be cleared of debris or bushes that could hinder the movement of the chainsaw operator [18,19].

Felling time is classified into two categories, namely effective time and delay time. Effective time consists of (a) preparing the tree to be felled, (b) felling, covers making undercuts and back cuts, and (c) bucking, which is cutting the base and end of the log, while the delay time consists of (a) walking time to get to the targeted tree to be felled, (b) break and lunchtime, and (c) time for chainsaw repair and maintenance. The measurement of working time is based on the beginning and end of each work cycle element in logging activities, as follows:

1. Preparation is calculated from the start of determining the direction of the fallen tree, cleaning the dirt stuck to the tree, cutting the liana, and securing the path for the feller.
2. Felling is calculated from the start of making undercuts and back cuts until the tree falls.
3. Bucking is calculated by cutting the base of the trunk, removing the remaining buttresses of the tree until cutting the top end of the trunk as close as possible to the first branch.
4. Walking between trees is counted from the first tree that has been cut down to the next target tree to be felled.
5. Rest and lunch breaks are counted from the start of taking a break until the end of lunch.
6. Chainsaw maintenance and repair is calculated from the feller sharpening the chainsaw, refilling fuel and oil, and replacing damaged spare parts until the chainsaw can function properly.

#### 2.3.2. Data Collection

In this study, the working time data referred to the working time of each element in the felling activities, including: (1) preparation of trees to be felled; (2) making cutting notches (consisting of undercuts and back cuts until the tree falls) (3) bucking (cutting the stem base, and its branch-free stem); (4) walk to the next targeted tree to be felled; (5) maintain and repair chainsaws; and (6) taking of breaks and lunch. All working times were recorded in minutes as a time unit. In addition, the diameter and length of the logs were measured within the bucking plots.

2.3.3. Data Analysis

The volume of log, productivity, and felling costs were calculated using the following formulas modified [20]:

The volume of log (V):

$$V = 1/4\pi \left(\frac{1/2(Dp + Du)}{100}\right)^2 L \quad (1)$$

where: V = volume of log ($m^3$); Dp = log base diameter (cm); Du = log end diameter (cm); L = length of log (m).

Felling productivity (P):

$$P = V/T \quad (2)$$

where: P = felling productivity ($m^3/h$); V = volume of log ($m^3$); T = felling time (hour).

Felling cost:

$$Tf = (Fc + Vc)/P \quad (3)$$

where: Tf = Total cost of felling (IDR/$m^3$); Fc = fixed cost (IDR/h); Vc = variable cost (IDR/h); P = felling productivity ($m^3/h$).

$$\text{Effective felling time (minutes)} = Pre + F + B \quad (4)$$

where: Pre = Preparation time (minutes), F = Felling time (minutes), B = Bucking time (minutes).

The fixed cost using a chainsaw was calculated using the FAO formula [21], while the variable cost consisted of the following components [20]:

1. Maintenance cost

Chainsaw maintenance costs were calculated based on the value of 10% of the chainsaw investment cost divided by total working hours for one year.

2. Repair cost

Repair costs were approached from interviews with chainsaw operators including replacement costs on damaged saw spare parts, especially chains and blades.

3. Fuel cost

Fuel cost was counted by calculating the amount of fuel used (liters) multiplied by the fuel price based on direct observation.

4. Lubricants cost

The cost of lubricating oil was approached by 10% of fuel consumption multiplied by the price of lubricating oil.

5. Labor wage

Labor wage refers to the total wage for all felling workers (chainsaw operators and helpers), calculated based on the local provincial minimum wage which is converted into wages per working time (USD/h).

The data were processed using Excel 2010 software and then statistically analyzed using PWSTAT version 23 to determine the distribution and proportion of felling working time concerning its influencing factors. To find out the relationship between the factors that affect the total time of felling, the regression equation was tested.

## 3. Results

### 3.1. Distribution of Working Time

The results of this study are presented in Table 1. Total felling time ranged from 15.3 to 18.3 min/tree with an average of 16.7 min/tree. The effective felling time was 11.8 min/tree (70.86%) and the delay time was 4.9 min/tree (29.14%).

**Table 1.** Resume of working time of felling in natural forest, West Papua, Indonesia.

| Work Element | Working Time per Tree | | | | | Effective Working Hours (%) |
|---|---|---|---|---|---|---|
| | Sampling Plot 1 | Sampling Plot 2 | Sampling Plot 3 | Average | | |
| | (Minutes) | (Minutes) | (Minutes) | (Minutes) | (%) | |
| A. Effective time | | | | | | |
| 1. Preparation | 6.3 | 5.3 | 6.1 | 5.9 | 35.33 | 49.86 |
| 2. Felling | 3.8 | 3.4 | 2.6 | 3.3 | 19.56 | 27.61 |
| - Under cut | 1.7 | 1.5 | 1.8 | 1.7 | 9.98 | 14.08 |
| - Back cut | 1.1 | 1.3 | 0.8 | 1.1 | 6.39 | 9.01 |
| 3. Bucking | 2.6 | 3.1 | 2.3 | 2.7 | 15.97 | 22.54 |
| - Cut the base of the stem | 1.5 | 2.2 | 1.3 | 1.7 | 9.89 | 13.96 |
| - Cut the end of the stem | 1.1 | 0.9 | 1.0 | 1.0 | 5.99 | 8.45 |
| Total (A) | 12.7 | 11.8 | 11.0 | 11.8 | 70.86 | 100.00 |
| B. Delay time | | | | | | |
| 1. Walking among tree | 1.7 | 2.0 | 1.1 | 1.6 | 9.58 | |
| 2. Rest and lunch break | 1.8 | 1.8 | 1.7 | 1.8 | 10.58 | |
| 3. Repairing and maintenancing chainsaw | 2.1 | 0.9 | 1.5 | 1.5 | 8.98 | |
| Total (B) | 5.6 | 4.7 | 4.3 | 4.9 | 29.14 | |
| Total felling time | 18.2 | 16.4 | 15.3 | 16.7 | 100.00 | |

Note: the number of trees felled on sampling plots 1, 2, and 3 were 16, 16, and 17 trees, respectively.

### 3.2. Productivity and Felling Cost

This study revealed that felling productivity was between 14.901 $m^3$/h and 16.221 $m^3$/h, or 15.153 $m^3$/h on average as shown in Table 2.

**Table 2.** Productivity and felling costs in West Papua, Indonesia.

| Variable | | Plot 1 | Plot 2 | Plot 3 | Average |
|---|---|---|---|---|---|
| Tree diameter | cm | 66.4 | 63.1 | 62.7 | 64.07 |
| Buttress height | cm | 172.9 | 147.1 | 151.6 | 157.22 |
| Slope | % | 25.5 | 21.3 | 18.8 | 21.84 |
| A. Felling productivity | $m^3$/h | 14.579 | 16.221 | 14.692 | 15.153 |
| 1. Log volume | $m^3$/tree | 4.43 | 4.437 | 3.748 | 4.205 |
| 2. Total felling time | minutes/tree | 18.2 | 16.4 | 15.3 | 16.6 |
| B. Chainsaw operating costs | | | | | |
| 1. Fixed cost | USD/h | 0.271 | 0.271 | 0.271 | 0.271 |
| 2. Varibale cost | USD/h | 0.931 | 0.839 | 0.824 | 0.865 |
| 3. Salary (*) | USD/h | 2.271 | 2.271 | 2.271 | 2.271 |
| Cost of felling | USD/h | 3.473 | 3.381 | 3.366 | 3.407 |
| | USD/$m^3$ | 0.238 | 0.209 | 0.229 | 0.225 |

Remark: (*) = Based on the assumption that the salary of the helper is equal to the minimum wage of West Papua. Province in 2022 of USD 237.037/month, and the wages of chainsaw operators are 30% higher than the helper.

To examine the correlation among tree diameter, buttress height, and land slope to felling time, a correlation analysis was performed using SPSS version 18, as presented in Table 3. Table 3 confirms that tree diameter and buttresses height significantly affected the total felling time.

**Table 3.** Analysis of correlation among diameter, buttress, and slope to felling time.

| Correlations | | | | | |
|---|---|---|---|---|---|
| | | **Diameters** | **Buttress** | **Slope** | **Felling Time** |
| Diameters | Pearson Correlation | 1 | 0.718 ** | 0.405 ** | 0.845 ** |
| | Sig. (2-tailed) | | 0.000 | 0.004 | 0.000 |
| | N | 49 | 49 | 49 | 49 |
| Buttress | Pearson Correlation | 0.718 ** | 1 | 0.417 ** | 0.655 ** |
| | Sig. (2-tailed) | 0.000 | | 0.003 | 0.000 |
| | N | 49 | 49 | 49 | 49 |
| Slope | Pearson Correlation | 0.405 ** | 0.417 ** | 1 | 0.426 ** |
| | Sig. (2-tailed) | 0.004 | 0.003 | | 0.002 |
| | N | 49 | 49 | 49 | 49 |
| Felling time | Pearson Correlation | 0.845 ** | 0.655 ** | 0.426 ** | 1 |
| | Sig. (2-tailed) | 0.000 | 0.000 | 0.002 | |
| | N | 49 | 49 | 49 | 49 |

**. Correlation is significant at the 0.01 level (2-tailed).

Based on Table 3, the coefficient of determination ($R^2$) clarifies that 72% of the total felling time was influenced by variables of tree diameter, buttress height, and slope. The rest (28%) was influenced by other variables, such as the skill of the saw operator, the distance between the trees being felled, the number and shape of buttresses, the type of tree (hardwood/softwood), and the skills of the helper.

Table 3 shows that the tree diameter, buttress height, and slope factors as a whole had a significant correlation with the total felling time.

## 4. Discussion

### *4.1. Distribution of Working Time*

Based on Table 1, these findings indicate that effective felling time is not optimal yet since chainsaw operators have not attended formal training on felling techniques; instead they only rely on the self-taught experience of felling trees when they were helpers in the past. Although currently forest concessions in felling are required to apply reduced-impact logging techniques, there is no regulation requiring chainsaw operators to have formal certification in logging education and training. To increase the effectiveness of felling time, the skills of operators need to be improved, through training in felling techniques among other methods. The felling activity is presented in Figure 2.

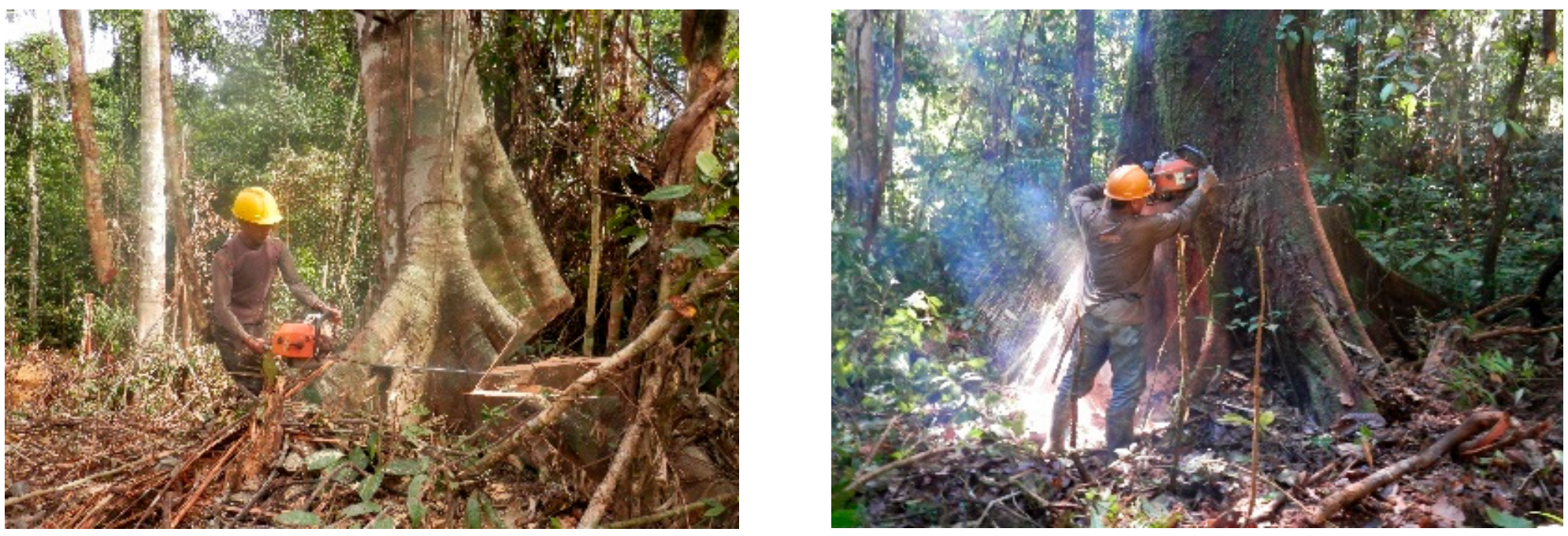

**Figure 2.** Inappropriate logging activities because chain saw operators have never been trained in reduced impact logging.

Compared to other scholars, this study finding has a longer time than [22], which is an average of 13.3 min/tree, as well as [23] which is an average of 14.6 min/tree, in the case of Central Kalimantan. Meanwhile, the total felling time in the Central Appalachian Hardwood Forest was of 12.77 min/tree [24]. Even the research finding from the Caspian hardwood forest in Northern Iran stated that the average total felling time of 4.57 min/tree

(delay time was excluded) [25]. Another finding from [26] in Northern Bosnia and Herzegovina indicated that felling time by an assortment method was 6.29 min/tree, while half tree length needed 4.16 min/tree. In addition, experience from the Carpathian Mountains, Romania, exemplifies a total felling time of 8.94 min/tree [27].

The longest time of felling in this study site was the preparation of trees to be cut, ranging from 5.3 min/tree to 6.1 min/tree with an average of 5.9 min/tree or 35.33%.

Table 1 indicates that making undercuts takes an average of 1.7 min/tree (9.98%) longer than making back cuts of 1.10 min/tree (6.39%). This is because there are at least 3 steps needed in making an undercut, namely (a) cutting the buttress, (b) making a tilted notch at an angle of approximately 45°, and (c) making the base of the notch in a horizontal cut to meets the part of the undercut. The higher buttress, the longer time to cut the tree. Therefore, undercuts and back cuts must be made correctly and carefully since it will determine the falling tree direction and wood quality, and ensure operators' safety. The difficulty in making undercuts and back cuts is if the tree is on a steep topography and has high buttresses. Consequently, the chainsaw operator should find the safest position, and the helper must be alert and ready to signal to the operator.

The study also found the delay time between the plots, ranging from 4.3 min/tree to 5.6 min/tree or 4.9 min/tree on average. The time lost is mostly due to repairing broken or pinched saw chains, and the process to find the targeted trees. The average working time to get the targeted tree of 1.6 min/tree (9.58%). It was highly influenced by stand density. In the case of the operator working without a tree distribution map, the delay time will be worse. Therefore, operators must be completed a detailed tree distribution map, and if possible, should also be accompanied by a planning officer who brings the forest inventory data and the Geographical Position satellite (GPS) tool.

Meanwhile, chainsaw maintenance and repair time varied considerably between the plots, ranging from 0.9 min/tree to 2.1 min/tree (or 1.5 min/tree on average). The difference came from a variety of engine disturbances, refueling, checking the saw, sharpening the chain, and overcoming obstacles due to the chainsaw being squeezed. Looking at the average proportion of walking time of the felled trees, which varies between the plots, this is rational, depending on the tree distance ranging from 25 to 125 m, micro topographic conditions with the hilly area, and slippery soil conditions due to wetness. Although the delay time is considered detrimental to the overall felling time, it plays an important role in maintaining the health of the operator and saw engine.

Table 3 shows that the tree diameter and buttress height factors have a correlation of 84.5% and 65.5%, respectively, higher than the slope factor of 42.6% to the total felling time. This means that the larger the diameter of the tree and the height of the buttress, the greater the effect on the total amount of time required for felling. This is easy to understand because the larger the diameter of the tree, the longer it takes to make undercuts and back cuts. Observations in the field the length of time for making undercuts and back cuts is also influenced by the wood hardness of the tree species. Likewise, the height of the buttress, the higher the buttress, the longer it will take to make undercuts and back cuts because they must first cut the buttresses of the tree. In addition to the height of the buttress, the number and shape are thought to affect the duration of undercut and back cut making. The low influence of slope on the total time of felling is thought to be due to the very small slope interval, which is around 6.7% of the slope of 18.8% to 25.5%.

Many factors affect the total felling time, among other tree distance, tree species, diameter breast height (dbh), tree buttress height, slope, and soil type in the felling area. However, this study indicated that tree diameter, buttress height, and slope significantly influence total felling time.

This study's finding is in line with other scholars. Tree diameter and distance among the trees significantly affect felling time [24,25]. Furthermore, the walking time between trees is strongly influenced by the density of the stand, especially the density of the mature tree to be felled [25]. The denser the mature trees, the shorter the walking time between the

trees. Felling time will get longer when increasing the tree diameter and distance of the felled trees, but it decreases with increasing air temperature [3].

### 4.2. Productivity and Felling Cost

The felling productivity found in this study is lower than other scholars' findings, such as [3,25,28,29] who reported 20.6 $m^3$/h; 44.61 $m^3$/h; 16.88 $m^3$/h; 30.08 $m^3$/h; and 32.80 $m^3$/h, respectively. However, Ref. [30] stated that felling productivity by chainsaws was from 10.138 $m^3$/h to 11.374 $m^3$/h and was strongly influenced by tree diameter. Meanwhile, the study conducted by [7] showed that the felling productivity of chainsaws with and without delay time was 56.4 $m^3$/h and 80.7 $m^3$/h, respectively. The skill of the operator is suspected to be the cause of the significant difference in felling productivity, assuming the same condition on the average tree volume and diameter, the distance among the trees, and the topography. Another influencing factor of felling productivity is the difference in seasons, where felling productivity was higher during summer (5.70 $m^3$/h) than winter (3.81 $m^3$/h) [12], stand and field conditions, type of equipment, management objectives, and operator experience [6,31].

Table 2 also shows that the felling costs ranged from 0.208 USD/$m^3$ to 0.238 USD/$m^3$ with an average of 0.225 USD/ $m^3$. Compared to another study conducted in tropical forests at Central Kalimantan, the felling cost is not significantly different, which ranged from 3.001 USD/$m^3$ to 0.504 USD/$m^3$ with an average of 0.374 USD/$m^3$ [20]. However, compared to another study in North Kalimantan, the felling cost was 0.084 USD/$m^3$ [28], and the felling costs found in this study were considered expensive. In addition, the cost of felling in this study is lower than the cost of felling in the Hyrcanian forest in Iran, which was reported to be 0.55 USD/$m^3$ with delay time, as well as lower than 0.39 USD/$m^3$ without delay time [7].

## 5. Conclusions

The total felling time ranged from 15.3 to 18.3 min/tree with an average of 16.7 min/tree, consisting of an effective felling time of 11.8 min/tree (70.86%) and a delay time of 4.9 min/tree (29.14%). Making undercuts took an average of 1.7 min/tree (9.98%) longer than making back cuts of 1.10 min/tree (6.39%). Total felling time was influenced by tree diameter, buttress height, and slope. The larger the tree's diameter, the height of the buttress, and the more sloping the field conditions, the longer the felling time. The effective cutting time is not yet considered optimal since the chainsaw operator has never received training in felling techniques and lacks experience in felling natural forest trees.

Felling productivity ranged from 14.579 $m^3$/h to 16.221 $m^3$/h, averaging 15.153 $m^3$/h. Differences in felling productivity are caused by slope, timber volume, the dominance of wood species, tree density, the experience of the felling team, and logging company policies. Felling costs ranged from 3.367 USD/h to 3.474 USD/h with an average of 3.407 USD/h or equivalent ranging from 0.208 USD/$m^3$ to 0.238 USD/$m^3$ with an average of 0.225 USD/$m^3$. This study implies the need to improve the effectiveness and productivity of felling time by the following approaches: (1) improve the skills of chainsaw operators through formal or in-house training on felling techniques, (2) require chainsaw operators to carry a tree distribution map completed by information on field conditions, and (3) ensure that operators are accompanied, if necessary, by a planning officer.

**Author Contributions:** Each author (S., D., Y., S.S., S.G., Y.R., A.S. and S.A.) had an equal role as the main contributor in which they equally discussed the conceptual ideas and the outline, provided critical feedback for each section, and helped shape and write the manuscript. All authors have read and agreed to the published version of the manuscript.

**Funding:** This research was funded by P.T. Wijaya Sentosa.

**Data Availability Statement:** All the data presented or analyzed during this study are included in the article.

**Acknowledgments:** The authors express high gratitude to P.T. Wijaya Sentosa for funding and facilitating the study.

**Conflicts of Interest:** The authors declare no conflict of interest.

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
