# Peer review of "Working Time, Productivity, and Cost of Felling in a Tropical Forest: A Case Study from Wijaya Sentosa’s Forest Concession Area, West Papua, Indonesia"

_forests, doi:10.3390/f13111789_

Round 1
Reviewer 1 Report
Table 1 should be corrected. Part A (TOTAL A) .the sum of 100 is a wrong calculation. how they got 100? from where?
similar to part B., please check the calculation of % ( check all element sampling plot time taken).
Section 3. results should add "s" Results
in method, please add a formula for Effective working hour
Table 1 How do they derive a percentage of the average? eg 35.33, 19.56,.....
Reviewer 2 Report
Line 16: Add an introductory sentence.
Line 18: The authors show the results before briefly describing the methodology. What were the species and minimum stem diameter? What was the average volume of these trees? Was it selective logging? Was a time study conducted?
What was considered as cutting? Was it only considered the operation time until the tree fell? It is not clear.
It would be interesting to say what influence factors were considered.
Convert IDR to USD.
Keywords: productivity and felling cost were used in the title. Remove and add other words.
Introduction
Line 59: "1,500 - 3,000 man-hours". The authors said it was per day. The sentence needs to be revised. Are the values for a group of men?
The importance of the study to the international community was not well described.
Materials and Methods
Figure 1 should be corrected. There is text in front of the figure. The legend is not visible.
Describe the characteristics of the forest and the species.
Describe the harvesting system and the characteristics of the chainsaws.
Describe the characteristics of the operators: age, number of operators evaluated, time of experience. Other information, if any.
Describe climate and soil. Methodology for measuring slope.
Describe the climatic conditions during data collection.
Line 115: The beginning and end of each element of the work cycle should be well defined. The terms used in the time study should be standardized according to IUFRO.
Formulas 1, 2 and 3 must be referenced.
Insert formula for the utilization rate. See: PULKKI, R. E. Forest Harvesting, I: On the Procurement of Wood with Emphasis on Boreal and Great Lakes St. Lawrence Forest Regions. 2001. 156 pp.
Calculate cost using dollar. FAO (1992) should be replaced with a more current methodology. An example:
Sessions, J., Michael, B., & Sup-Han, H. (2021). Machine rate estimates and equipment utilization-A modified approach. Croatian Journal of Forest Engineering: Journal for Theory and Application of Forestry Engineering, 42(3), 437-443.
In the wage values it was not described if social contributions were considered.
The sampling error was not described. Use Murphy's formula (2005)
MURPHY, G. Determining sample size for harvesting cost estimation. New Zealand Journal of Forestry Science, v. 35, n. 2/3, p. 166-169, 2005.
The description of the statistical analysis is not complete. Describe the correlation analysis. Describe which variables were considered.
Line 163-169: The text is methodology. It is not result.
Note table 1 should be described in the methodology. See Murphy's calculation and state the sampling error of the study.
Line 173: Text repeats information from line 162. Delete repeated sentences.
Table 2: Describe in the methodology how the slope of the land was obtained.
Replace IDR for Dollar. State in the methodology whether hours are with or without delays. Use PMH instead of hour to say productive machine hours.
Line 180: The text looks like discussion. Move text and reference.
Line 196: State in the methodology the regression equations tested.
Figure 2: Presenting a correlation analysis is more appropriate than a trend line.
Line 205: Isn't there a labor law that requires formal training?
It would be interesting to have a photo of the study site, to better understand its characteristics. I can't imagine what the forest is like and the posture adopted by the chainsaw operators. After all, they had no training.
Line 221: It's methodology. Move text. Figure 3 is methodology.
Line 266: Remove equation from text. Already said in the results.
Remove from text: (Wang & McNeel 2004; Behjou et al. 2009). (Behjou et al. 2009).
Reviewer 3 Report
Minor English spell check required (lines 204, 247). Your statistical model has 3 predictors, and you conclude that they influence 72% of the total felling time. You speculate that skill of the chainsaw operator should be improved to improve the effectiveness and productivity of felling time. Yet, you didn't include any operator skill metric in your analysis or in the Materials and Methods section (such as years of experience). Comparing obtained felling costs with different regions (Iran) is not necessary due to significant socioeconomic differences. The paper is well written, and it provides good data for tropical forest logging, yet it lacks originality.
Round 2
Reviewer 2 Report
Version 2 is different from trackchange.
Line 186: Replace IDR for USD.
Figure 2 is missing. Figure 3 in trackchange version is different.
Line 225: The description is methodology. Describe the analysis and software in the last paragraph of the methodology.
